# AN EXPLAINABLE AI-BASED COMPLEMENTARY ATTENTION MECHANISM FOR DETECTING IDENTITY SWAPS

## ABSTRACT

Deep learning techniques have quickly led to the generation of a large number of realistic fake content by accessing large-scale publicly available databases. The emergence of deepfake technology has given rise to concerns related to the creation and dissemination of manipulated multimedia content because of its use in social media to generate fake news. One prevalent application of this technology is identity swap, wherein faces are exchanged within images and videos to create convincing yet fabricated visual narratives. Thus, the detection of identity swaps has become an increasingly important research area in the field of digital forensics. This paper presents a complementary attention-based deep learning system for the detection of identity swaps. Specifically, it incorporates our proposed simple Layer-Integrated Channel Attention (LICA) and Scaled Spatial Attention (SSA) mechanisms in the VGG network architecture to respectively capture the importance along each channel and at each spatial location to distinguish real faces from manipulated faces. It further incorporates Local Interpretable Model-agnostic Explanations (LIME) as the explainable AI technique to provide a more in-depth transparent analysis of its effectiveness towards improved detection performance. Our extensive experimental results demonstrate that the proposed system outperforms state-of-the-art systems in terms of accuracy and area under curve metrics in detecting fake faces generated by identity swaps. The LIME further provides a deeper understanding of the decision-making process and facilitates trust and accountability by combining the power of CNNs with the transparency of explainable AI.

## 1 INTRODUCTION

The emergence of manipulated and fake images has raised concerns regarding the authenticity and integrity of visual content (Dang et al., 2019). The ability to quickly and accurately detect and distinguish between real and fake images has become critical in various research areas including computer vision, image forensics, and multimedia security. Particularly, the advancements in digital editing tools and deep learning techniques have made it easier than ever to create realistic yet manipulated facial images. These manipulated facial images can be used for malicious purposes such as spreading misinformation, identity theft, and creating deceptive content in various domains including journalism, social media, biometric systems, and law enforcement. Consequently, there is a pressing need for robust and efficient methods to identify and classify manipulated faces accurately.

However, the detection of fake faces poses several challenges due to the increasing sophistication of manipulation techniques, which range from simple image retouching to more complex methods like face swapping, expression manipulation, and deepfake synthesis. In addition, these manipulation techniques often aim to deceive human perception by altering facial attributes, expressions, or even entire identities.

Therefore, developing effective approaches for fake face detection has become an active and evolving research area.

Early approaches to fake face detection relied on handcrafted features and rule-based algorithms (Dang et al., 2019) to analyze pixel inconsistencies or geometric anomalies in facial images. However, handcrafted features do not have the capability of defining specific features, especially for face representation. As a result, the detection performance is limited. Recently, the availability of large-scale datasets such as Celeb-DF (Li et al., 2020), FaceForensics++ (Rössler et al., 2019), and the DeepFake Detection Challenge (DFDC), which includes diverse samples of real and manipulated faces in different realistic and challenging scenarios, has facilitated the development and evaluation of deep-learning-based detection algorithms. Specifically, Convolutional Neural Networks (CNNs) based methods (Li et al., 2020) (Kolagati et al., 2022) leverage the power of deep neural networks to automatically learn discriminative features to differentiate between real and manipulated faces and demonstrate promising results.

In this paper, we present a novel approach for the detection of fake faces using advanced deep learning techniques. Our proposed method leverages a simple channel and spatial attention, namely Layer-Integrated Channel Attention (LICA) and Scaled Spatial Attention (SSA), in a VGG19-based deep learning architecture to differentiate between real and manipulated faces. The paper further incorporates Local Interpretable Model-agnostic Explanations (LIME) as the explainable AI technique to provide a more in-depth analysis of decision making of the models used. We conduct extensive experiments on the DFFD dataset (Dang et al., 2020), specifically designed for fake face detection, to evaluate the performance and effectiveness of our approach. The remainder of this paper is organized as follows: Section 2 reviews the related work in fake face detection. Section 3 presents the framework and methodology of the proposed approach. Section 4 describes the dataset used for evaluation. Section 5 introduces the experimental setup and presents the results and performance analysis. Finally, Section 6 concludes the paper and highlights the contributions and potential future directions in fake face detection.

## 2 RELATED WORKS

In this section, we briefly review some representative deep-learning-based fake face detection methods. Li et al. (2020) utilize a pre-trained CNN to extract deep features from images and train a classifier to identify manipulated faces based on learned representations. Kolagati et al. (2022) propose to combine facial landmarks including eye, lip, nose shape, and eye blink detection with deep learning to detect manipulated faces. Hsu et al. (2018)introduce a Generative Adversarial Network (GAN) based approach that incorporates a discriminator network to detect manipulated faces by exploiting inconsistencies in facial structures. Current research efforts focus on developing more robust and generalizable methods that can handle various synthesis methods, address adversarial attacks, and consider temporal information for video-based fake face detection.

FaceSwap (Kowalski, 2018) and DeepFakes (Korshunov & Marcel, 2018) are common techniques to generate identity swap faces. We review a few representative techniques to detect fake face generated by identify swap techniques. Nguyen et al. (2019) proposed an autoencoder-based detection system that achieved an Equal Error Rate (EER) of 15.07% on fake faces, in the FaceForensics++ database (Rössler et al., 2019), that are generated by the FaceSwap manipulation method. However, this model does not generalize very well for fake images generated by other manipulation techniques and achieves an Area Under Curve (AUC) of less than 80%. Recently, attention models have been widely used in various computer vision tasks including image classification (Wang et al., 2017), image inpainting (Caicedo & Lazebnik, 2015), and object detection (Yoo et al., 2015). To the best of our knowledge, only one pioneer work by Li et al. (2020) has been developed to use the attention mechanism for face manipulation detection and localization. It incorporates an attention mechanism in a CNN-based detection system to improve the training process to learn more effective feature maps to represent facial features. This attention-based model is tested on the DFFD database (Dang et al., 2019), which combines the FaceForensics++ database (Rössler et al., 2019) and a collection of videos from the Internet. It achieves an AUC of 99.43% and an EER of 3.1% and improves the non-attention-based CNN detection system.

The explainability of a machine learning model has become significantly important. As a result, the field of explainable AI has raised in popularity. Barredo Arrieta et al. (2020) discusses the importance of explainable AI and presents various approaches and challenges in achieving transparency and interpretability in AI systems. Explainable AI in CNNs has been a significant area of research.

In order to make CNN models more interpretable and transparent a number of explainable AI techniques were proposed over time. Selvaraju et al. (2017) introduces Grad-CAM (Gradient-weighted Class Activation Mapping), a technique to generate visual explanations for CNN predictions. It highlights the importance of localization and provides insights into what parts of an image influenced the model's decision. LIME: Local Interpretable Model-Agnostic Explanations (Ribeiro et al., 2016) is a popular method for generating model-agnostic explanations for machine learning models, including CNNs.

## 3 METHODOLOGY

We have designed a deep learning-based technique to detect identity swaps. We use the state-of-the-art CNN model VGG19 (Simonyan & Zisserman, 2015) as a backbone to detect manipulated face images. On top of that we introduce two simple yet effective attention mechanisms, namely, Layer-Integrated Channel Attention (LICA) and Scaled Spatial Attention (SSA), and incorporate them into the VGG19 network to achieve better detection performance in terms of accuracy and AUC. Our proposed method leverages deep learning architectures and attention mechanisms to differentiate between real and manipulated faces. Finally, we use LIME (Ribeiro et al., 2016) for interpreting our model's decision making. LIME which stands for Local Interpretable Model-Agnostic Explanations is a framework and technique in the field of explainable AI.

### 3.1 NETWORK ARCHITECTURE

We use the VGG19 network architecture shown in Fig. 1 to build the CNN. It offers several advantages in the task of identifying fake images. 1) The deep and wide architecture allows it to learn more complex features and recognize intricate patterns from the input data. 2) The VGG19 model has already been pre-trained on a large dataset, which makes it more computationally efficient compared to training a model from scratch. 3) The VGG19 model is capable of being fine-tuned with real and fake face images to achieve better performance.

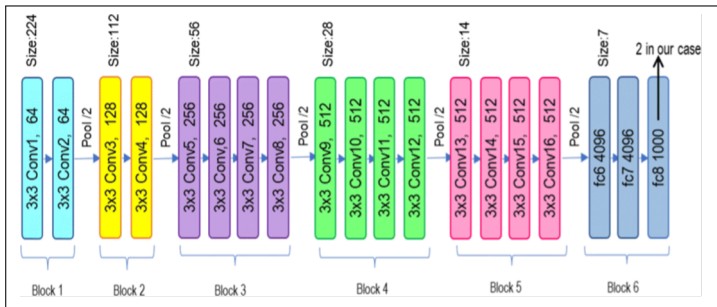

Figure 1: VGG19 Network Architecture (Khattar & Quadri, 2022)

### 3.2 LAYER-INTEGRATED CHANNEL ATTENTION (LICA) AND SCALED SPATIAL ATTENTION (SSA)

We propose two attentions and incorporate them after the last dense convolutional layer (or before fully connected layers). The first attention is called LICA and the second attention is called SSA. LICA is followed by SSA to capture the importance along each channel and at each spatial location.

LICA repeatedly combines the data across channels to learn the relative importance of data across the channels. Fig. 2 presents the proposed LICA, where $Input \epsilon R^{C \times H \times W}$ with $C$ representing the number of channels, $H$ representing the number of rows in each channel, and $W$ representing the number of columns in each channel. Below are the equations to compute each block in Fig. 2.

$$X_{\text{conv}} = \text{1x1 Conv(Input)} \tag{1}$$

$$Y_{\text{conv}} = \text{1x1 Conv(Input)} \tag{2}$$

$$\text{Relative} = S(\text{1x1 Conv}(X_{\text{Conv}} + Y_{\text{Conv}})) \tag{3}$$

$$\text{Result} = X_{\text{Conv}} + (X_{\text{Conv}} \times \text{Relative}) \tag{4}$$

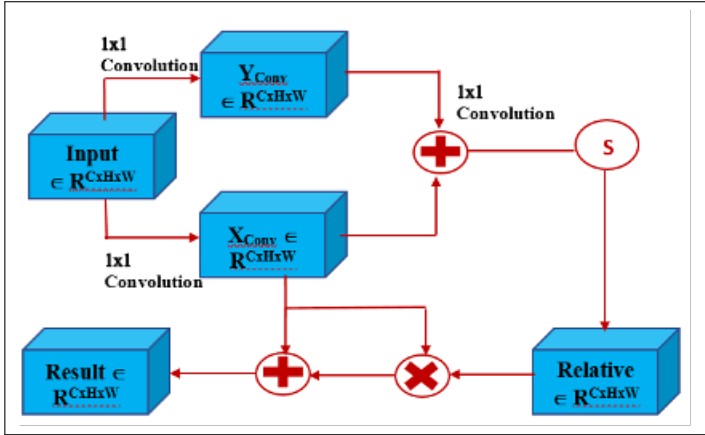

Figure 2: Illustration of the LICA, where the operators "+" "×", and "S" within the circle represent elementwise addition, elementwise multiplication, and sigmoid operations, respectively.

SSA plays an important role in deciding where to focus. Fig. 3 presents the proposed SSA. Below are the equations to compute each block in Fig. 3.

$$\text{Max} = \text{Maxpooling}(\text{Input}) \tag{5}$$

$$\text{Average} = \text{Averagepooling}(\text{Input}) \tag{6}$$

$$\text{Combined} = \text{Concatenate}(\text{Max}, textAverage) \tag{7}$$

$$\text{Scaled} = \text{Softmax}(\text{Convolutional}(\text{Combined})) \tag{8}$$

$$\text{Result} = \text{Input} + (\text{Input} \times \text{Scaled}) \tag{9}$$

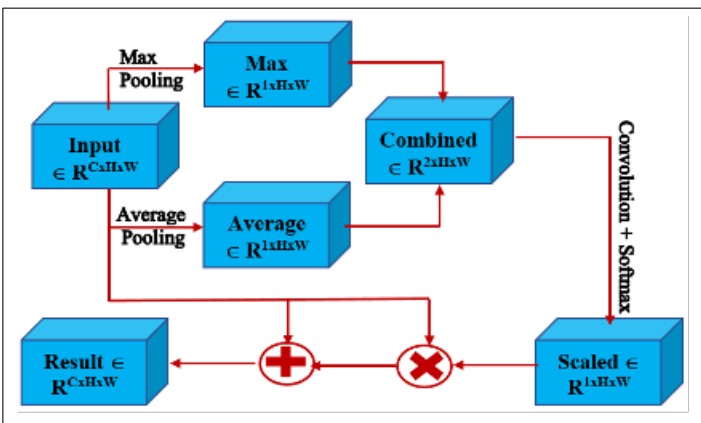

Figure 3: Illustration of the LICA, where the operators "+" "×", and "S" within the circle represent elementwise addition, elementwise multiplication, and sigmoid operations, respectively.

To efficiently compute SSA, max-pooling and average-pooling are individually applied to the input data to capture the maximum and average information of each input pixel in all channels. The two pooling results (e.g., Max and Average shown in Fig. 3) are then concatenated and passed through a convolution layer to squeeze all the information into one channel. The softmax operation is employed to obtain a scaled attention matrix (Scaled), where the sum of information in the channel is 1. This scaled attention matrix is then multiplied with the input data, which can be the original input or the output of channel attention, to capture the weighted spatial matrix, which is finally added back to the input data to get the final refined output. We use addition instead of multiplication to add more weighted information, which can be useful as descriptors later for the classification task.

## 4 DATASET

We have used the DFFD dataset (Li et al., 2020) for our training and evaluation. To achieve better accuracy, we use FaceNet's (Schroff et al., 2015) MTCNN to detect and crop the faces out from the frames extracted from the videos. The DFFD dataset contains four different types of manipulations. For our experiment, we focus on the identity swaps generated using DeepFake (Korshunov & Marcel, 2018) and Faceswap (Kowalski, 2018) to replace the face of one person in a video with the face of another person. We also use the Celeb-DF to test our models generalizability.

## 5 EXPERIMENTAL SETUP AND RESULTS

In order to have a balanced training set we chose to have the same number of manipulated images of each type. Also, the two number of manipulated images and real images were also kept same. Table I below shows the dataset split.

Table 1: Dataset Split

| Dataset | Train | Validation | Test |
|---|---|---|---|
| FF++(Original) | 2500 | 1000 | 2500 |
| DeepFake(Original) | 2500 | 1000 | 2500 |
| DeepFake | 2500 | 1000 | 2500 |
| FaceSwap | 2500 | 1000 | 2500 |

The detection results are summarized in Table II below.

Table 2: DETECTION RESULTS ON IDENTITY SWAP

| Model | Accuracy | AUC |
|---|---|---|
| VGG19 | 96.86% | 99.48% |
| VGG19 + complementary attention | 97.28% | 99.58% |
| VGG19 + attention in (Ling et al., 2020) | 96.78% | 99.37% |
| XceptionNet + regression map (Dang et al., 2020) | 96.23% | 99.24% |

The above shows that the VGG19 plus the proposed complementary attention achieves the highest accuracy and AUC when compared to other systems. The proposed detection system improves XceptionNet-based system by 1.05% in accuracy and 0.34% in AUC. We also show our proposed system's accuracy on each category of identity swap attack in the table below.

Table 3: DETECTION RESULTS ON EACH IDENTITY SWAP CATEGORY

| Category | Accuracy |
|---|---|
| DeepFake | 97.68% |
| FaceSwap | 93.08% |

### 5.1 GENERALIZABILITY

In order to test our model's generalizability we use the Celeb-DF dataset.We use MTCNN to detect and crop the faces out from the frames extracted from the videos in the dataset. We test our model (VGG19 plus complimentary attention)on 7865 such images to get an accuracy of 74.84%.

### 5.2 MODEL EXPLAINABILITY

In this section, we evaluate the explainability of our CNN based complementary attention system. LIME has been used to mark the superpixels that have contributed towards the classification label generated for a particular subject. The regions shown in green color have contributed positively for the predicted label and the regions shown in red color have contributed negatively to the predicted label.

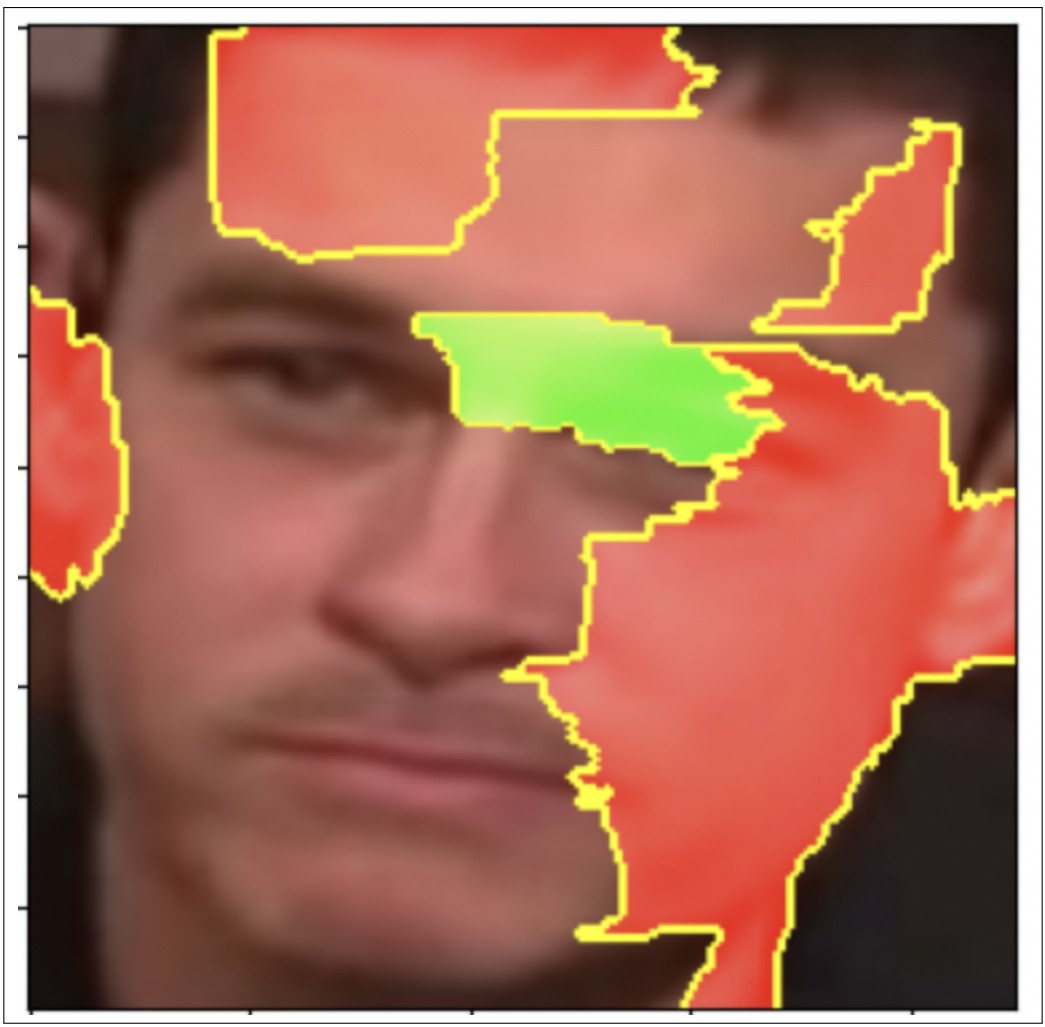

Figure 4: Celeb-DF fake image : Predicted "FAKE", Prediction Score "99.20%"

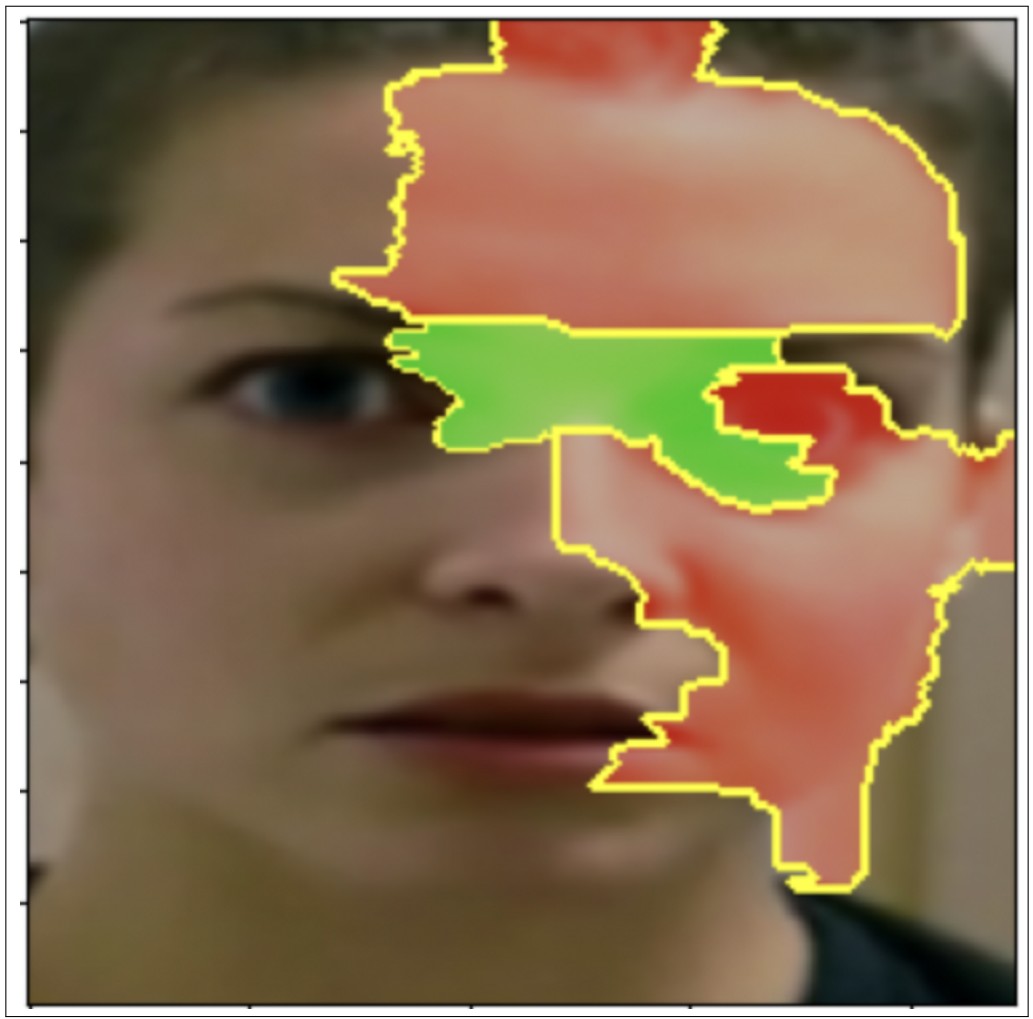

Figure 5: FF++ Deepfake image : Predicted "FAKE", Prediction Score "96.80%"

As we can observe in Fig. 4 and Fig 5, the model is successfully able to detect deepfakes from different datasets. The regions between the eyes and some part of the eyes play a positive impact in that decision. Other parts, like cheeks, forehead etc play a negative role in the prediction in both the cases. This tells us that our model gives importance to region between the eyes and some part of the eyes to determine if its a fake or not.

# 6 CONCLUSION

We propose two kinds of simple yet effective complementary attention, namely LICA and SSA, which can be combined or individually used in any deep-learning network architecture, to improve fake face detection performance in terms of accuracy and AUC. The experimental results demonstrate that the proposed system achieves an accuracy of 97.28% and an average AUC of 99.58% on fake faces generated by ID swap and outperforms its variants and the state-of-the-art system and its counterpart variants. The proposed system performs better than the other compared systems and also generalizes well across different datasets. With help of LIME we could further understand the regions in the image which were important for our model decision making. For future work, we will explore the possibility of adding residuals to further improve face detection results.

### ACKNOWLEDGMENTS

We would like to thank Michigan State University's Computer Vision Lab for the access to their DFFD dataset which was used in our research.

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
