# OpenReview forum: "An Explainable AI-based Complementary Attention Mechanism for Detecting Identity Swaps"
_ICLR.cc/2024/Conference — Submitted to ICLR 2024_

### Official Review · Reviewer_X4Gr · 2023-10-30

**Soundness:** 1 poor
**Presentation:** 1 poor
**Contribution:** 1 poor
**Rating:** 1
**Confidence:** 5

**Summary:**

In this paper, the authors propose adding an extra 1x1 convolutional kernel and a simple softmax layer to VGG-19 for detecting deepfakes. To put it succinctly, this is not a novel idea. Furthermore, there are significant issues with both the text and figures. As a result, the reviewer votes to 'reject.'

**Strengths:**

This paper is easy-to-follow.

**Weaknesses:**

1. The method lacks novelty. The proposed LICA and SSA are both simple modifications to the backbone model. Functionally, they resemble fine-tuning, transferring a pre-trained VGG-19 to the task of deepfake detection. This approach is essentially common sense in computer vision and cannot be considered an independent contribution.

2. The experiments lack comparisons with state-of-the-art (SOTA) methods. Deepfake detection sees numerous related works every year, and the authors do not compare with all of them. At the very least, it is expected that the authors compare their method with common benchmark methods such as Face X-ray, MultiAtt, RECCE, and others.

3. The manuscript requires more polishing. For instance, Section 1 devotes a substantial amount of space to discussing the necessity of this work. In Section 2, the summary of related work is limited to publications up to 2020, without summarizing or mentioning any recent developments.

4. All images have extremely low resolutions.

**Questions:**

n/a

---

### Official Review · Reviewer_Bbmb · 2023-10-30

**Soundness:** 3 good
**Presentation:** 2 fair
**Contribution:** 2 fair
**Rating:** 5
**Confidence:** 3

**Summary:**

The paper discusses the development of a complementary attention-based deep learning system for detecting identity swaps in fake images.  The proposed method incorporates Layer-Integrated Channel Attention (LICA) and Scaled Spatial Attention (SSA) mechanisms in the VGG network architecture to capture the importance along each channel and at each spatial location.

**Strengths:**

The topic could be of wide interest. Layer-Integrated Channel Attention and Scaled Spatial Attention mechanisms are proposed and sound reasonable.

**Weaknesses:**

The paper evaluates the proposed method on a limited number of deepfake models. It would be beneficial to include more diverse models to validate of the approach.

Besides, there are major problems in the writing and formatting of this article. 1) page 6 and page 7 are filled with oversized pictures. 2) It remains unclear to me the relation between the existing issues and the motivation of the proposed methodology, plus how it resolves the issues (in the current version, nor was it explicitly explained in the abstract or in the introduction).

**Questions:**

See weaknesses.

---

### Official Review · Reviewer_9aRj · 2023-11-08

**Soundness:** 2 fair
**Presentation:** 3 good
**Contribution:** 2 fair
**Rating:** 3
**Confidence:** 4

**Summary:**

This paper presents a complementary attention-based deep learning system for detecting identity swaps in fake face images. The proposed system incorporates Layer-Integrated Channel Attention (LICA) and Scaled Spatial Attention (SSA) mechanisms in the VGG network architecture to capture the importance along each channel and at each spatial location. It further utilizes Local Interpretable Model-agnostic Explanations (LIME) as an explainable AI technique to provide a transparent analysis of its effectiveness. Experimental results demonstrate that the proposed system outperforms state-of-the-art methods in terms of accuracy and area under curve metrics.

**Strengths:**

1. The Layer-Integrated Channel Attention (LICA) and Scaled Spatial Attention (SSA) mechanisms capture the importance along each channel and at each spatial location, enhancing the detection performance.

2. The proposed system outperforms state-of-the-art systems in terms of accuracy and area under curve metrics in detecting fake faces generated by identity swaps.

**Weaknesses:**

1. The evaluation of the proposed system is limited to the DFFD dataset, which may not fully represent the diversity and complexity of real-world scenarios.
2. In section 5.1, “We test our model (VGG19 plus complimentary attention)on 7865 such images to get an accuracy of 74.84%.” Can you give more evidence about this result?
3. This paper does not provide insights into the generalizability of the proposed approach to different types of manipulation techniques or datasets.
4. My comprehension of the operational utility of the Local Interpretable Model-Agnostic Explanations (LIME) method is currently lacking. In the context of the present research, LIME is exclusively deployed for the purpose of visualizing the model's outcomes. However, it is noteworthy that attention mechanisms, specifically the employment of an attention heatmap, are viable alternatives capable of achieving a similar objective. Can you explain why you chose LIME instead of attention heatmap?

**Questions:**

Can you explain why Layer-Integrated Channel Attention (LICA) can learn the relative importance of data across the channels?

---

### Meta-Review · Area_Chair_qk5B · 2023-12-06

**Metareview:**

The paper presents an attention-based deep network for detecting identity swaps in fake images. The evaluations are limited to the DFFD dataset, raising concerns about the method's generalizability and real-world applicability. There are also issues with the manuscript's clarity and presentation and a lack of comparison with recent related works. No rebuttal is provided.

**Justification For Why Not Higher Score:**

The paper's limited evaluation scope, lack of generalizability across diverse datasets, and issues in presentation and manuscript clarity prevent a higher score. The method's novelty is also questioned due to its resemblance to common techniques used.

**Justification For Why Not Lower Score:**

N/A

---

### Decision · Program_Chairs · 2024-01-16

Reject